# Detection and Characterization of a Reassortant *Mammalian Orthoreovirus* Isolated from Bats in Xinjiang, China

**DOI:** 10.3390/v14091897

**Published:** 2022-08-27

**Authors:** Xiaomin Yan, Jinliang Sheng, Chang Zhang, Nan Li, Le Yi, Zihan Zhao, Ye Feng, Changchun Tu, Biao He

**Affiliations:** 1Changchun Veterinary Research Institute, Chinese Academy of Agricultural Sciences, Changchun 130122, China; 2College of Animal Science and Technology, Shihezi University, Shihezi 832003, China; 3Jiangsu Co-innovation Center for Prevention and Control of Important Animal Infectious Diseases and Zoonosis, Yangzhou University, Yangzhou 225009, China

**Keywords:** *Mammalian orthoreovirus*, bat, isolation, pathogenicity, reassortment

## Abstract

*Mammalian orth**oreoviruses* (MRVs) are increasingly reported to cause various diseases in humans and other animals, with many possibly originating from bats, highlighting the urgent need to investigate the diversity of bat-borne MRVs (BtMRVs). Here, we report the detection and characterization of a reassortant MRV that was isolated from a bat colony in Xinjiang, China. The BtMRV showed a wide host and organ tropism and can efficiently propagate the cell lines of different animals. It caused mild damage in the lungs of the experimentally inoculated suckling mice and was able to replicate in multiple organs for up to three weeks post-inoculation. Complete genome analyses showed that the virus was closely related to MRVs in a wide range of animals. An intricate reassortment network was revealed between the BtMRV and MRVs of human, deer, cattle, civet and other bat species. Specifically, we found a bat-specific clade of segment M1 that provides a gene source for the reassortment of human MRVs. These data provide important insights to understand the diversity of MRVs and their natural circulation between bats, humans, and other animals. Further investigation and surveillance of MRV in bats and other animals are needed to control and prevent potential MRV-related diseases.

## 1. Introduction

Orthoreoviruses are a group of non-enveloped, segmented and double-stranded RNA viruses within the genus *Orthoreovirus* of the family *Reoviridae*. Currently, they are divided into ten species with *Mammalian orthoreovirus* as the prototype [1]. The virions are 70–80 nm in diameter with a typical icosahedral, double-layered protein capsid structure [2]. Their genomes are approximately 23.5 kb in length and contain three large (L1–L3), three medium (M1–M3), and four small segments (S1–S4), encoding eight structural (λ1, λ2, λ3, μ1, μ2, σ1, σ2 and σ3) and four nonstructural proteins (μNS, μNSC, σNS and σ1s) [3].

Orthoreoviruses are commonly identified in animals and some of them are potential causative agents of neurological disease, pneumonia, respiratory illness and enteritis in a variety of animals including humans [4,5,6,7]. Orthoreovirus infection is also likely to cause diarrhea in domestic animals as a number of *Mammalian orthoreoviruses* (MRVs) have been isolated from diarrheal pigs in China, South Korea and the United States [8,9,10]. In particular, two MRV3 strains isolated from diarrheal piglets showed high pathogenicity causing severe diarrhea and acute gastroenteritis, with mortality rates as high as 100% [11]. Therefore, orthoreoviruses are of increasing concern in public health and veterinary medicine.

Bats are considered important natural hosts for hundreds of viruses, including many important zoonotic pathogens such as the Marburg virus, Hendra virus and Lyssavirus [12,13,14]. Due to their diversity, wide distribution and long-distance flight, bats are prone to disperse viruses to various animals over a wide area [15], e.g., our previous investigation revealed that bat-borne rotaviruses contributed many genomic segments to human-, horse- and rabbit rotaviruses [16,17]. Bats are also reported to carry multiple orthoreoviruses, suggesting that the creatures also serve as potential natural reservoirs of the viruses [18,19,20,21]. Notably, a bat-borne orthoreovirus, the Melaka virus, was identified to cause acute respiratory disease in humans, indicating that bat-borne orthoreoviruses can be transmitted to and cause disease in humans [7]. Thus, it is necessary to investigate the diversity of bat-borne orthoreoviruses so as to understand their short- and long-term effects in terms of public health.

Here, we report the detection and characterization of a new reassortant bat-borne *Mammalian orthoreovirus* (BtMRV) isolated from a bat colony in Xinjiang, China. The virus was prevalent in the intestine and lung tissues of the bat population and exhibited a close genetic relationship with MRVs from humans, deer, civets and cattle. The virus could efficiently replicate in cell lines of several animal species and showed mild pathogenicity in experimentally infected mice. These data provide important insights to understand the diversity of MRVs and their natural circulation between bats, humans and other animals.

## 2. Materials and Methods

### 2.1. Sample Information and Virus Detection

In 2016, 46 and 76 apparently healthy *Pipistrellus pipistrellus* bats were sampled at abandoned houses in Xinyuan and Qapqal counties in Ili Prefecture, Xinjiang Uygur Autonomous Region, China. The two counties were ~170 km apart. Lungs and intestines from sampled bats have previously been used for viral metagenomic analysis and the detection of parechovirus and polyomavirus [22,23]. The previous viromic data revealed MRV-like sequences [24]. A semi-nested RT-PCR method was designed to screen all samples in order to validate the result. The primer sequences were designed based on the assembled contigs using SPAdes. Total RNA of each sample was extracted using QIAamp RNeasy Mini Kit (Qiagen, Hilden, Germany). Reverse transcription was performed using the commercial 1st cDNA synthesis kit (TaKaRa, Dalian, China). The first round of RT-PCR was performed with the forward primer 5′-GGATGGATGTATCAGGGAA-3′ and the reverse primer 5’- CCATACCACCTAAAC AGCAG-3’, and the second round of PCR was conducted using a combination of the same forward primer as used in the first-round and the reverse primer 5′-GGTAGAAG GTTGTTCATC-3′. The PCR reaction was carried out using a PCR master mix (Tiangen, Beijing, China) with a cycling condition of 35 cycles of denaturation at 94 °C for 30 s, annealing at 56 °C for 30 s, and extension at 72 °C for 40 s. Double-distilled water was used as a negative control.

### 2.2. Virus Isolation and Cell Tropism Test

Virus isolation was initially carried out using the porcine kidney 15 (PK-15, ATCC CCL -33) cell line. Eleven MRV-positive intestines were individually homogenized with serum-free Dulbecco’s modified Eagle medium (DMEM) (HyClone, Logan, UT, USA) and were centrifuged at 12,000× *g* for 10 min at 4 °C. The supernatants were passed through 0.22-µm filters and then were added to cells for 1 h incubation. After discarding the supernatants, cells were maintained in DMEM supplemented with 2% fetal bovine serum (FBS) and a 2-fold dose of 100,000 U/mL penicillin and 10,000 g/mL streptomycin (Invitrogen, Carlsbad, CA, USA). The infected cells were inspected daily for cytopathogenic effect (CPE). The indirect immunofluorescence assay (IFA) we previously established to detect bat rotavirus [16] was modified to validate isolation. We prepared antiviral hyperimmune sera by performing the intramuscular inoculation of adult mice with the purified virions. The secondary antibodies in the IFA were Alexa Fluor 488-conjugated donkey anti-mouse IgG(H+L) (Invitrogen, Carlsbad, CA, USA), and the Spearman–Karber method was used to calculate the 50% tissue culture infectious dose (TCID_50_) of the virus.

Cell tropism of the virus was assessed using PK15, African green monkey kidney (Vero, ATCC CCL -81), baby hamster kidney-21 (BHK-21, ATCC CCL-10), Madin-Darby bovine kidney (MDBK, ATCC CCL-22), human cervical carcinoma cell (HeLa, ATCC CCL-2) and an immortalized embryonic *Myotis petax* bat kidney cell line (BFK) established by our laboratory [25]. After inoculation, cells were observed daily for CPE, and the titer in each cell line (except for PK-15) at 72 h post-inoculation (h.p.i.) at passage 3 was determined using IFA. 

For electron microscopy, the PK-15 culture at 72 h.p.i. at passage 6 was harvested by three cycles of freezing-thawing, followed by centrifugation at 12,000× *g* for 5 min. The supernatants were passed through a 0.22 μm filter, and 10 μL filtrates were negatively stained with 2% phosphotungstic acid and examined with a JEM-1200 EXII transmission electron microscope (TEM) (JEOL, Tokyo, Japan). In addition, the PK-15 culture showing 50% CPE was scraped to prepare the ultrathin section. After centrifugation at 3000× *g* for 15 min at 4 ℃, cells were fixed overnight using 2.5% glutaraldehyde, and then using 1% OsO_4_ for 2 h. Fixed cells were dehydrated using acetone for 1 h and were impregnated with an acetone-epoxy resin mixture. After embedding, samples were polymerized at 60 ℃, cut into ultrathin sections, and stained using a mixture of 1% uranyl acetate and 1% lead citrate. Each slice was observed using a TEMH-7650 TEM (Hitachi, Tokyo, Japan).

### 2.3. Genome Sequencing and Phylogenetic Analysis

To obtain the complete genomic sequence of the virus, we performed RNAseq of the three isolates at passage 6 on an Illumina NovaSeq sequencer using our MTT protocol [26]. The clean data were de novo assembled using SPAdes, and the contigs were annotated using BLASTx search against our eukaryotic viral reference database (EVRD) [27] with an e-value cut-off of 10^−5^. The open reading frames (ORFs) were predicted using the ORF finder [28]. The resulting nucleotide (nt) and amino acid (aa) sequences of putative viral proteins were further validated by performing an online BLASTn/x search against the non-redundant database of Genbank. The clean reads were mapped and backed to the obtained genomic sequences using Bowtie2 to estimate the sequencing depth and coverage. Reference sequences of the 10 segments were retrieved from Genbank. The multiple sequence alignments (MSAs) were achieved using ClustalW of MEGA X [29]. The phylogeny of each segment was reconstructed using the maximum likelihood (ML) method with the best nucleotide substitution model which was assessed using the Akaike information criterion implemented in Model Finder. Branch support was assessed using the bootstrap analysis of 1000 replicates. The resulting trees were visualized and polished with Interactive Tree of Life (iTOL) [30].

### 2.4. Reassortment Analysis

Potential genomic reassortment was initially identified based on the phylogenetic topology of the 10 segments. The sequences of the isolate QAPpC66 in this study and the potential reassortment parents were used for further validation. We trimmed the untranslated regions (UTRs) at both termini of each segment and concatenated them in the order of L1-L2-L3-M1-M2-M3-S1-S2-S3-S4. The MSA of tandem constructs was generated using ClustalW and then subjected to reassortment prediction using RDP v.4.95 with the algorithms of RDP, GENECONV, Chimaera, MaxChi, BootScan and SiScan [31]. The reassortment event was considered significant if it was supported by at least three algorithms (*p*-values < 10^−10^). Reassortment was further visualized using SimPlot v.3.5.1 [32].

### 2.5. Experimental Infection

Pregnant Kunming mice (*n* = 5) were purchased from Liaoning Changsheng Biotechnology Co., Ltd, Shenyang, China. Three-day-old suckling mice were randomly divided into four groups with twelve animals in each. Two groups were either orally or intracranially inoculated with 3 × 10^5^ TCID_50_ of the isolate QAPpC66. The other two control groups were mock-inoculated with uninfected PK-15 culture in the same way. Animals were weighed daily and monitored for clinical signs. Three animals in each group were euthanized at 7, 14, 21 and 28 days post-inoculation (d.p.i.) and were subject to examination of the organ lesions. The rectum, lung, liver, brain and kidney of each animal were collected for MRV detection. A quantitative RT-PCR (RT-qPCR) was designed to detect and quantify the genome copy of QAPpC66. The primer and probe sequences were Reo-QF1 (5′-ATCTTATAAGCGCGTGCCGA-3′), Reo-QR1 (5′-GGAGATATCCATGCAC GCCA-3′) and Reo-probe (FAM5′-TACACTTGATGGTGATACGGTCC-3′ TAMRA). These tissues were accurately weighed and completely ground before RNA extraction. The RT-qPCR program was performed using the MonAmp Taqman qPCR mix (Monad, Wuhan, China) under the following program: 15 s at 95 °C, then 40 cycles of 5 s at 95 °C, 30 s at 60 °C.

### 2.6. Statistical Analysis

All statistical analyses were performed using SPSS 22.0 software (IBM, Armonk, NY, USA) [33]. Values *p* < 0.05 were considered statistically significant.

## 3. Results

### 3.1. Detection of MRV RNA in Bats

Based on our previous viromic results [24], we designed a semi-nest RT-PCR targeting a contig that was annotated to be an MRV M2 fragment. The brain, lung, liver, intestine, and kidney samples of the 122 *Pipistrellus pipistrellus* bats were subjected to RT-PCR detection. The results revealed that the prevalence of MRV was higher in Qapqal (26.3%, 20/76) than in Xinyuan county (10.9%, 5/46) (Table 1). For Qapqal, the virus was detected in the five tissue types with a positive rate of 2.6–14.5%, while the virus was only detected in a few liver, lung, and intestine tissues with a prevalence of 2.1–6.5% in Xinyuan. All 213-bp amplicons were sequenced, and sequence comparison revealed that they were 99.5–100% nt identical to each other, suggesting that these strains were close variants of a single MRV. The naming format of these variants contains information including location (the first two letters), bat species (the second two letters), tissue type (the last letter) and a number (sampling order).

### 3.2. In Vitro Isolation and Characterization of MRV

Eleven MRV-positive intestine samples from Qapqal were subjected to isolation using PK-15 and three of them induced obvious CPEs from passage 2. RT-PCR detection verified the successful isolation of three MRV isolates (QAPpC11, 15 and 66), while the remaining samples were negative after five passages. Compared to mock-infected cells (Figure 1A), MRV-infected PK-15 showed granulation and shrinkage, and >90% of cells detached at 72 h.p.i. (Figure 1B). A wide range of cell lines from human, monkey, hamster, cattle and bat (*Myotis petax*) were used to test the infectivity of the MRV. All these cells are susceptible and permissible to the virus and showed similar CPEs (Figure 1B–G). The three isolates of passage 6 showed identical replication dynamics in PK-15 and reached a peak titer of 10^6.67–7^ TCID_50_/mL at 48 h.p.i. (Figure 1H). The replication ability of isolate QAPpC66 in other cell lines was evaluated for passage 3, resulting in higher titers (10^6.00−6.83^ TCID_50_/mL) in BHK, Vero, BFK and Hela cell lines, while its replication in the MDBK cell line (10^4.67^ TCID_50_/mL) was less efficient (Figure 1I).

Electron microscopy of the isolate QAPpC66 in PK-15 at passage 6 showed three morphotypes of virus particles (Figure 2A). The mature virions were typical orthoreovirus-like with a size of ~80 nm in diameter. Infectious subvirion particles (ISVPs) lacking σ3 proteins [34], which are slightly smaller than those of typical virions, were also detected. In addition, many core particles were present with a lack of outer-capsid proteins including μ1, σ1 and σ3, making them less infective and much smaller compared to the other particle forms [35]. The ultrathin sections showed that a large number of electron-dense virus particles were arrayed in a paracrystalline form and visible in the cytoplasm of the infected PK-15 cell (Figure 2B).

### 3.3. Results of Experimental Infection

To assess pathogenicity, the isolate QAPpC66 was orally and intracranially inoculated into suckling mice. Generally, these inoculated mice did not show overt clinical signs throughout the observation with a few animals at 2–5 d.p.i. demonstrating slight depression. The necropsy of animals at 7 d.p.i. showed that one mouse in the intracranial group (IG) had needle-like hemorrhages in the lung (data not shown), while two of the three mice in the oral group (OG) demonstrated thinner intestinal walls than those of the control group. qRT-PCR detection showed that the virus was positive in the brain, lung and kidney tissues of one of three IG animals with virus loads of 1.9–4.5 × 10^3^ genome copies/g at 7 d.p.i., and were detectable until 14 d.p.i. with virus loads decreased to 9 × 10^2^–2.1 × 10^3^ genome copies/g in some liver and brain tissues. The virus was able to replicate in the brain and lung tissues of two of three OG mice with virus loads of 3.9–4.0 × 10^3^ copies/g at 7 d.p.i., and the replication in lung continued until 21 d.p.i. with a virus load of 4.1 × 10^2^ genome copies/g.

### 3.4. Full Genome Characterization and Phylogenetic Analyses

Isolates QAPpC11, 15 and 66 were all subjected to RNAseq. After de novo assembly and blastx annotation, contigs corresponding to the 10 segmented genes were successfully obtained for the three isolates. They were 23,268 bp, 23,424 bp and 23,352 bp in length, all covering the CDS regions. Nevertheless, the UTRs of 5’ and 3’ ends could not be determined. They were 99.7–100% similar to each other at nt level with the same lengths of encoding genes, indicating that they were isolates of the same virus. Comparison with other known reference sequences available in GenBank showed that these segments were closely related to MRVs originating from human, cattle, deer, bat and civet, with as high as 98.2% nt similarities (Table 2, Figure 3 and Figure 4), and the three isolates fell into MRV serotype 2 based on the phylogeny of S1 (Figure 4). These segments showed distinct phylogenetic topologies and clustered with different genetic neighbors (Figure 3 and Figure 4). As for segment L1 which encodes RNA-dependent RNA polymerase, the three isolates were next to YNSZ/V207 with ~97% nt similarity, representing a cattle MRV detected in the Chinese Yunnan province, 2016 (Figure 3). For L2, L3 and S2, our isolates formed a clade with a human MRV (Figure 3 and Figure 4), which was reported to cause diarrhea of a Slovenian traveler who returned from Southeast Asia in 2017 [36]. Regarding M1 which encodes a component of major outer-capsid protein, the three isolates neighbored four European MRVs (Figure 3), three of which originated from bats in Germany [37], Italy [38] and Slovenia [39], respectively, while the remaining one was recovered from human stool in Switzerland [40]. Interestingly, for M2, M3, S1, S3 and S4, the three isolates clustered with MRVs of a wide range of hosts, such as a bat and a civet in China, and a deer in the USA (Figure 3 and Figure 4). These results elucidated the intricate reassortment network of MRV.

### 3.5. Reassortment Analysis

To reveal reassortment events among these viruses, a comprehensive reassortment analysis was conducted using BtMRV-QAPpC66 as a query. Several reassortment events were detected in the junction regions of these segments (Figure 5A). Consistent with the phylogenetic analyses, six viruses were predicted to be parental strains (Figure 5A). Three Chinese MRVs, i.e., YNSZ/V207 from cattle [36], MPC/04 from a civet [41] and WIV4 from a bat [42], were involved in the reassortment of L1, S3, M2 and S4 with QAPpC66 (Figure 5B). Potential reassortments of M3 and S1 of QAPpC66 were observed with the American deer MRV OV204 [43] (Figure 5B). Notably, mew716 from a Swiss child [40] participated in the reassortment of M1 with QAPpC66 (Figure 5B), while the Slovenian SI-MRV07 reassorted with QAPpC66 in segments L2, L3 and S2 (Figure 5B).

## 4. Discussion

Orthoreoviruses have been increasingly detected in bats worldwide, with many of them capable of infecting humans and other animals [5,44,45]. Here, we successfully isolated three strains of a BtMRV that showed a close genetic relationship with MRVs of various origins. The BtMRV was detected in two bat colonies, with higher prevalence in Qapqal than in Xinyuan country. The virus has a broad tissue tropism and can be detected in multiple organs of these bats, which was also verified by its efficient replication in different cell lines as well as by the experimental infection. Of note is that the positive rate in the intestine samples was significantly higher than in other organs. BtMRV was successfully isolated from three intestinal samples, and the animal infection experiment revealed that oral infection can lead to intestinal abnormality. These results suggest that the fecal–oral route is likely a main transmission path of the virus. The pathogenicity of BtMRVs has been rarely evaluated. Jiang et al. investigated the pathogenicity of three BtMRVs and found that they are highly pathogenic to BALB/c mice and can induce systematic infection with obvious tissue damage in the lungs and intestines [46]. The three BtMRVs were isolated from insectivorous bats in Southern China [42] and showed high similarities in segments M2, M3, and S4 with the virus we isolated here (Figure 3 and Figure 4). Our experimental infection with the isolate QAPpC66 also caused mild lesion formation and hemorrhage in certain lungs of Kunming mice, and was replicated in multiple organs at up to 21 d.p.i. These data suggest that BtMRVs are of pathogenic potential, and their public health threats merit further investigation. Notably, our animal infections did not show overt pathogenicity to suckling mice compared to those revealed by Jiang et al., which could be due to the different mouse strains used. However, another reason should be noted which is that the pathogenic strain used by Jiang et al. belongs to serotype 3. Viruses within this serotype also showed high mortality in American piglets [11], which might suggest that the pathogenicity of MRV varies in serotype. 

The virus isolated here is closely genetically related to other bat-borne MRVs with as high as ~98% nt similarities in segments M2, M3, and S4 (Table 2), but it is less similar to MRVs of other animal origins for the remaining segments (Table 2). Besides, these BtMRVs fell into different phylloclades in almost all segments, e.g., they were classified into three serotypes based on segment S1 (Figure 3 and Figure 4). These indicate that BtMRVs are not only genetically divergent, but also rich in genetic diversity, which should be concerning as BtMRVs are important sources for the reassortment of other animal-origin MRVs and cross-host transmission, given the long-distance flight and wide distribution of bats.

The phylogenetic and reassortment analyses revealed an intricate reassortment network of MRVs among bats, humans and other animals. However, we realized that some reassortments did not occur recently, and even required other intermediate events. For example, the reassortment of S1 was predicted to occur between QAPpC66 and an American deer MRV strain OV24, but the two segments were just 92.4% identical and were from two different continents (Figure 4 and Table 2). Thus, the long-term evolution of the segment is required to bridge the genetic gap, which might involve other intermediate viruses. Of note is that the MRV isolated from a Slovenian traveler with gastroenteritis who returned from Southeast Asia [36] showed similarities as high as ~95% nt with isolate QAPpC66 in as many as three segments (Figure 3 and Figure 4), which indicates multiple reassortments of MRVs between bats and humans. However, it is difficult to determine the direction of gene flow in the reassortment event. Previously, we successfully disentangled the interspecies transmission and reassortment of rotaviruses between bats and humans or other animals. In this study, although multiple BtMRVs were involved in the analyses, almost all of their gene segments clustered with those of other animals, i.e., the segments of BtMRVs did not form a specific clade, but rather exchanged frequently with the viruses of other origins. Surprisingly, we noted that a clade in segment M1-based phylogeny was almost comprised of BtMRVs, consisting only of our three isolates and three European BtMRVs as well as a human MRV. The three European viruses were isolated from three insectivorous bat species from Germany, Italy and Slovenia between 2008 and 2011 [37,38,39]. Such an evolutionary convergence of M1 indicates that the clades are likely bat-specific and its members are widely distributed in Eurasian insectivorous bats. Conversely, the M1 of the Swiss child strain mew716 was most likely reassorted from this bat-specific clade, which illustrates a concerning phenomenon that BtMRVs provide a reassortment source for human viruses.

In summary, these BtMRVs have extensive host tropisms, making them easily spread to different animal species by multiple transmission routes. With the advantage of their genome segmentation, they can exchange genomic segments as the co-infection of multiple viruses occurs in a cell or a host, resulting in a new reassortant [36,39,47]. Moreover, the spillover of BtMRVs into humans and other animals has been increasingly documented, with many of them responsible for different diseases in recent years [36,44,46,47]. This highlights the urgent need for the continued investigation and surveillance of MRVs, not only in bats but also in other animals, which will improve our understanding of the diversity and circulation of MRVs, as well as help us to take proactive measures to control and prevent MRV-related diseases.

## Figures and Tables

**Figure 1 viruses-14-01897-f001:**
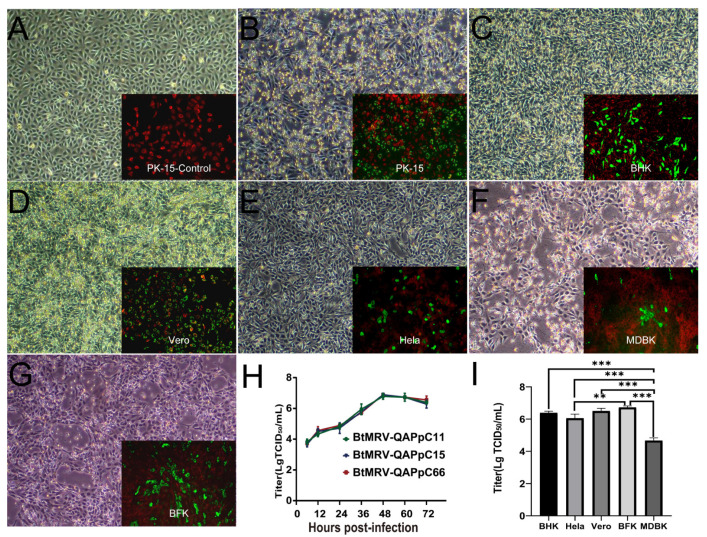
Infectivity of the BtMRV in different cell lines. (**A**) Mock-infected PK-15 cells show a normal appearance with negative detection by IFA. (**B**–**G**) CPE induced by BtMRV-QAPpC66 in PK-15, BHK, Vero, Hela, MDBK, and BFK cells with infected cells showing positive detection (bright apple green fluorescence) by IFA (inlet). (**H**) The replication dynamics of QAPpC11, 15 and 66 in PK-15 at passage 6. (**I**) The titers of QAPpC66 at passage 3 in five cell lines at 72 h.p.i. (**, *p* < 0.01; ***, *p* < 0.001; both by Bonferroni’s multiple comparisons test).

**Figure 2 viruses-14-01897-f002:**
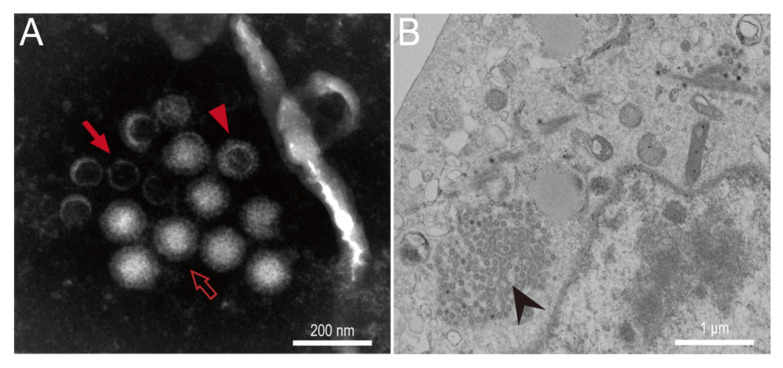
Electron microscopy of BtMRV-QAPpC66. (**A**) Three morphotypes of BtMRV-QAPpC66, i.e., typical virions (open arrow), infectious subvirions (filled triangle) and core particles (filled arrow) can be observed under TEM. (**B**) Ultrathin section of the infected PK-15 cell shows arrayed virions (arrowed) in cytoplasm.

**Figure 3 viruses-14-01897-f003:**
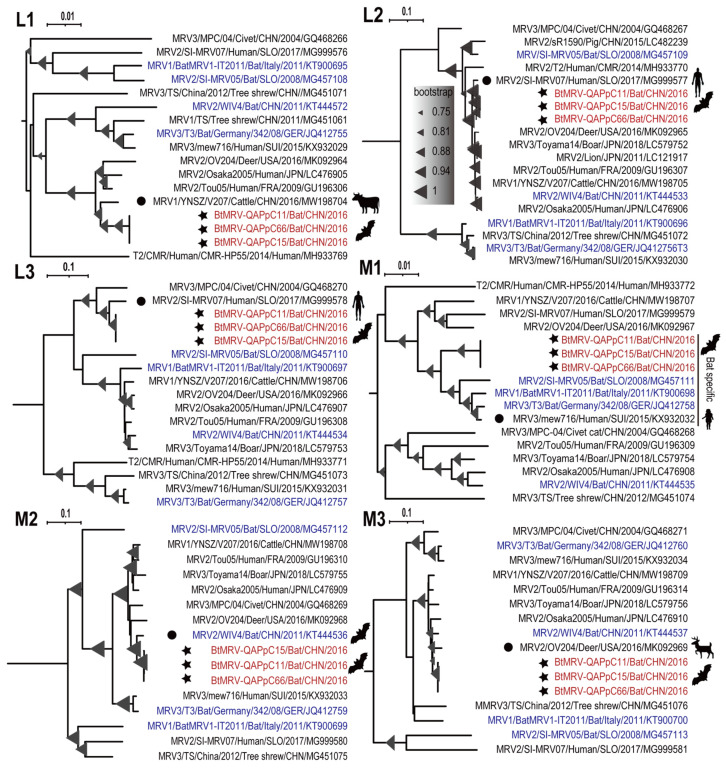
Phylogenetic analyses of the (**L1**–**M3**) segments based on nt sequences. The phylogenetic reconstruction was implemented using the maximum likelihood method and the best-fit model with 1000 bootstrap replicates. Each MRV is identified with its serotype, name, host, country, year, and GenBank accession number. Bootstrap value greater than 75 is shown on the node as a triangle with its size indicating the value (shown in panel L2). The three bat MRV isolates in this study are headed with filled stars, and their closest genetic neighbors are indicated with filled circles. The scale bar indicates the genetic distance. The animal cartoons indicate the hosts of the viruses they are next to, and their meanings are shown in Figure 5.

**Figure 4 viruses-14-01897-f004:**
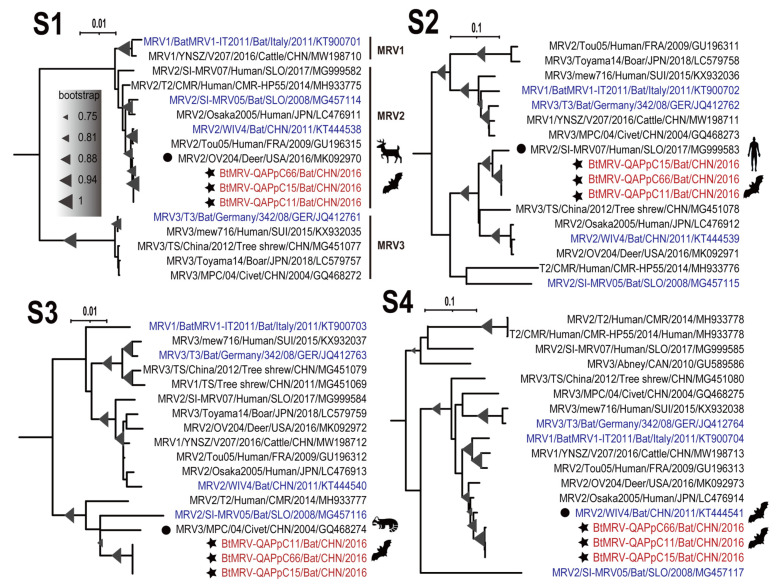
Phylogenetic analyses of the (**S1**–**S4**) segments based on nt sequences. The phylogeny of S1 indicates that the three isolates belong to MRV serotype 2. The phylogenetic reconstruction was implemented using the maximum likelihood method and applied the best-fit model with 1000 bootstrap replicates. Each MRV strain is identified with its serotype, name, host, country, year, and GenBank accession number. Bootstrap value greater than 75 is shown on the node as a triangle with its size indicating the value (shown in panel S1). The three bat MRV isolates in this study are headed with filled stars, and the closest genetic neighbors are indicated with filled circles. The scale bar indicates the genetic distance. The animal cartoons indicate the hosts of the viruses they are next to, and their meanings are shown in Figure 5.

**Figure 5 viruses-14-01897-f005:**
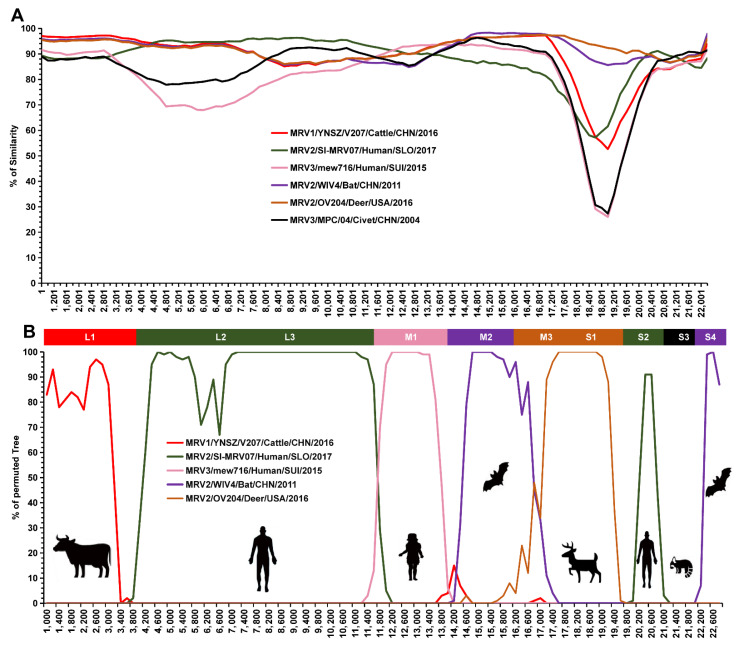
Reassortment analysis of BtMRV-QAPpC66 and other MRVs. (**A**) Similarity plots and (**B**) bootscan analyses were performed with the concatenated segments of BtMRV-QAPpC66 as the query and other MRVs, including MRV1/YNSZ/V207/Cattle/CHN/2016, MRV2/SI-MRV07/Human /SLO/2017, MRV3/mew716/Human/SUI/2015, MRV2/WIV4/Bat/CHN/2011, MRV2/OV204/Deer/ USA/2016 and MRV3/MPC/04/Civet/CHN/2004, as potential parental sequences.

**Table 1 viruses-14-01897-t001:** RT-PCR detection of MRV in bat samples.

County	Tissues					Individuals
	Brains	Livers	Kidneys	Lungs	Intestines	
Qapqal	7.9% (6/76)	2.6% (2/76)	7.9% (6/76)	9.2% (7/76)	14.5% (11/76)	26.3% (20/76)
Xinyuan	0% (0/46)	6.5% (3/46)	0% (0/46)	2.1% (1/46)	4.3% (2/46)	10.9% (5/46)

**Table 2 viruses-14-01897-t002:** The highest nt similarities of BtMRV QAPpC11, 15 and 66 to their reference sequences. NA: not available.

Segments	Similarity	Strains	Host	Disease	Country	Years	Accession No.
L1	97.0%	YNSZ/V207/2016	Cattle	NA	China	2016	MW198704.1
L2	95.1%	SI-MRV07	Human	Diarrhea	Slovenia	2017	MG999577.1
L3	93.6–95.9%	SI-MRV07	Human	Diarrhea	Slovenia	2017	MG999578.1
M1	93.8%	mew716	Human	Flu-like, diarrhea	Switzerland	2015	KX932032.1
M2	98.0–98.1%	WIV4	Bat.	NA	China	2011	KT444536.1
M3	98.1–98.2%	SI-MRV04	Bat	NA	Slovenia	2009	MG457103.1
S1	92.4%	OV204	Deer	Lethargy	USA	2016	MK092970.1
S2	95.7–95.9%	SI-MRV07	Human	Diarrhea	Slovenia	2017	MG999583.1
S3	91.2–91.3%	MPC/04	Civet	Diarrhea	China	2004	GQ468274.1
S4	97.8–97.9%	WIV5	Bat	NA	China	2011	KT444551.1

## Data Availability

The complete genomes of the three viruses were deposited in GenBank under accession numbers OP169447-OP169476. The RNAseq raw data were available in the NCBI Sequence Read Archive (SRA) under accession numbers SRR20930584-SRR20930586.

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
