# Peer review of "Detection and Characterization of a Reassortant Mammalian Orthoreovirus Isolated from Bats in Xinjiang, China"

_viruses, 2022, doi:10.3390/v14091897_

Round 1

Reviewer 1 Report

Yan et al have sequenced and analyzed a group of bat-borne mammalian orthreovirus (BtMRVs) from Pipistrellus pipistrellus in China. 25 BtMRV-positive individuals were detected and 3 of them were isolated successful in several mammalian cell lines. One isolate was proved the pathogenicity by experiment on mice and found multiple-organ pathology. Through a variety of phylogenetic and sequence-based analyses, this predicts a series of genomic reassortment event. I think this study is an effort of great public health significance and should be published after a few details modified.

1. Figure 5 should have two aligned abscissa axis, same type size, and same font color.

2. Line 210, 211, and 212, reference genes should be considered when the research need to compare RNA genome numbers from different tissues, because each tissues have different cell type and inhomogeneous distribution. I suggest that either this section should be increased, or presented with much more details to indicate complete tissues were fully ground and lysed.

3. Line 131, does phylogenetic analysis of each segment use a same nucleotide substitution model K2P? If it is, calculating nucleotide substitution model should be performed respectively.

4. The manuscript requires proof-reading to unify some of the terms used. Line 137, 142, and 143: the use of “recombine and reassort” should be verified.; Line 67, what's the meaning of “,and.”?; The word usage about times of passages should be unified and normalized.

Reviewer 2 Report

Dear authors!

Thank you for preparing such an interesting article on BtMRVs! I have minor comments on your paper listed in the attachment.

Best regards!
